# Isolation and Characterisation of Cellulose Nanofibre and Lignin from Oil Palm Empty Fruit Bunches

**DOI:** 10.3390/ma13102290

**Published:** 2020-05-15

**Authors:** Saharman Gea, Amir Hamzah Siregar, Emma Zaidar, Mahyuni Harahap, Denny Pratama Indrawan, Yurika Almanda Perangin-Angin

**Affiliations:** Department of Chemistry, Faculty of Mathematics and Natural Sciences, Universitas Sumatera Utara, Jalan Bioteknologi No.1., Medan 20155, Indonesia; siregar_amirhamzah@yahoo.com (A.H.S.); ema3@usu.ac.id (E.Z.); harahap.mahyuni@gmail.com (M.H.); dennypratama96@gmail.com (D.P.I.); yurikaalmanda183@gmail.com (Y.A.P.-A.)

**Keywords:** cellulose nanofibre, lignin, palm empty fruit bunch, steam explosion, soda-pulping

## Abstract

A study on isolation and characterisation of cellulose nanofibre (CNF) and lignin was conducted to expand the application of CNF and lignin from oil palm biomass. CNF was extracted by steam explosion and the by-product was precipitated to obtain lignin by using the soda-pulping method. The concentrations of NaOH used for CNF by-product precipitation were 2%, 4%, and 6%. The morphology of CNF and lignin was characterised using scanning electron microscopy (SEM). The nanofibre of CNF with dimension between 50 nm and 100 nm was investigated using transmission electron microscopy (TEM). The functional group was observed using Fourier-transform infrared (FTIR) spectroscopy, showing that CNF had the structure of cellulose-I. In addition, the chemical structures of isolated and commercial lignin were analysed using ^1^H-NMR spectrometry. CNF had a 72% crystallinity index characterised by X-ray diffraction (XRD), while lignin showed an amorphous form. The characterisation of isolated lignin was compared with commercial lignin. The two lignins had similar particle size distribution from 1 to 100 μm. From UV-visible analysis, the lignin had aromatic rings/non-conjugated phenolic groups. The morphology of isolated lignin was rough and flaky. Commercial lignin was in powder form with near-spherical morphology. Thermogravimetric analysis (TGA) of CNF showed 30% of residue at 600 °C. The results showed a simple method to isolate CNF and lignin from oil palm empty fruit bunches.

## 1. Introduction

The synthesis of fine chemicals and bio-based functional materials has attracted huge attention and created significant public value with the increasing of environmental and ecological concerns due to the use of petroleum-based chemicals and products [1]. Cellulose nanofibre (CNF) is extracted from cellulose (the most abundant natural polymer in the world). It has desirable properties such as low density, non-toxicity, and biodegradability. Additionally, it has unique properties such as high mechanical strength, reinforcement capabilities, and tunable self-assembly in aqueous media, arising from its unique shape, size, surface chemistry, and high degree of crystallinity [2].

CNF has a Young’s modulus between 20 and 50 GPa with surface areas of hundreds of square meters per gram [3]. These properties cause the biopolymer to have many new promising properties and applications [4,5]. Scientists have started to produce innovative materials from CNF for novel and emerging applications to tackle environmental problems originating from an abundance of renewable biomass. Several studies have reported the isolation of CNF from biomass, such as CNF from bagasse using acid and ball milling method [6], from tomato peels using acidified sodium chloride [7], and from a banana peel using high ultrasonication and chemical treatment [8]. The biomass can provide an excellent alternative to petroleum and other valuable commodities. CNF is used to produce a variety of high-value products with low environmental and societal impact [9]. It has been reported that the presence of CNF in a polymer matrix improves the mechanical properties due to the large specific surface area and high aspect ratio. For example, Balea et al. reported that CNF improved mechanical recycled paper properties by 15.1% with the addition of 3 wt% CNF [10]. In addition, CNF isolated from corn organosolv pulp increased the tensile index of recycled paper by 20% [11]. 

Lignin is the most dominant aromatic polymer on Earth and the second most abundant natural resource after cellulose [12]. It comprises up to 20% to 30% of the weight of woody plants [13,14]. Lignin, a by-product of pulp and paper industry, is globally produced at approximately 70 million tons annually [15]. This polymer has been developed for valuable applications such as macromolecular toughening agents for epoxy resin, surfactants for nanomaterials, and carbon nanosheets for high-capacitance supercapacitors. Lignin is considered as an ideal carbon fibre precursor due to its availability in nature, affordability, and bituminous coal-like structure [16]. It has been reported that a Kraft hardwood lignin and an organic-purified hardwood lignin have potential as precursors for low-cost carbon fibre. Lignin-based carbon fibre has a tensile strength of 0.51 GPa and a tensile modulus of 28.6 GPa [17]. In addition, carbon fibre from acetylated lignin has a tensile modulus, strength, and strain-to-failure values of 52 ± 2 GPa, 1.04 ± 0.10 GPa, and 2.0 ± 0.2%, respectively [18].

Carbon fibre consists of approximately 92% carbon, which is obtained through carbonisation. It is a lightweight material, has low density, and has good mechanical properties (stronger than steel and aluminum). The demand of carbon fibre increased significantly from 33 million tons in 2010 to 64 million tons in 2016, and it is projected to increase to 120 million tons in 2022 [19]. This is due to the increase in the use of carbon fibre, starting from sport equipment, aircraft, and advanced engineering application. Nowadays, the precursor of carbon fibre has been developed from natural resources because of environmental concern. Lignin-based carbon fibre is currently in the “Research and Development” stage. Some studies have reported the usage of lignin as a carbon fibre precursor. For example, Zhang and Zhao carbonised lignin to produce three-dimensional porous carbon [20], while Mainka reported the use of lignin as an alternative precursor for sustainable automotive carbon [21]. However, the Young’s modulus and strength of the carbon fibre is still low. It is expected that by blending CNF and lignin, the mechanical properties of lignin-based carbon fibre can be increased. In addition, the production cost of lignin-based carbon fibre is estimated to be about $4–5/lb, compared to that of petroleum-based carbon fibre, $10/lb. at commercial scale [22]. 

In this study, we are interested in isolating lignin and CNF from palm empty fruit bunches (PEFB) with a simple method (the flowchart is presented in Figure 1). PEFB consist of cellulose (65%), lignin (29.2%), hemicellulose (28.8%), and extractive substances (3.7%). There are many studies that reported the utilisation of PEFB-based cellulose, for example, pulp production, activated carbon, and potassium fertiliser for oil palm planting [23]. However, lignin derived from by-product is still less utilised. In this study, we isolated CNF by steam explosion and lignin was extracted from the by-product through the soda-pulping precipitation method. Lignin and CNF of PEFB have the potential to be used as precursors for carbon fibre and reinforcement materials.

## 2. Materials and Methods 

### 2.1. Materials

In this study, palm empty fruit bunches (PEFB) were used as a raw material to produce cellulose nanofibre (CNF) and lignin. PEFB were obtained from palm plantation managed by Universitas Sumatera Utara. Some chemical reagents used such as NaOH, CH_3_COOH, NaOCl, H_2_O_2_, HCl, H_2_SO_4_ were purchased from Merck (Damstadt, Germany). 

### 2.2. Isolation and Purification of Cellulose Nanofibre

Cellulose nanofibre (CNF) was isolated by steam explosion based on our previous work [24]. Firstly, PEFB fibre was cut into small pieces and immersed in 2% NaOH for 24 h. The suspension was then filtered. The residue was autoclaved at 168.9 kPa pressure in 130 °C for 4 h to collect CNF, and the filtrate was kept to isolate the lignin. The residue collected was neutralised with deionised water. After that, the sample was washed with 17.5% NaOH:7.4% CH_3_COOH:1.75% NaOCl solution with a ratio of 1:1:6. The fibre was bleached using 10% H_2_O_2_ solution and neutralised with deionised water until the pH was close to 7. The bleached fibre was hydrolysed using 10% HCl and ultrasonicated for 3 h at room temperature. Finally, CNF obtained was washed with deionised water and homogenised for 15 min at 8000 rpm at room temperature until the fibre was suspended. The suspension was filtered and CNF was dried in a vacuum oven at 50 °C for 4 h. The drying sample was coded as CNF.

### 2.3. Isolation and Purification of Lignin

NaOH was used to immerse PEFB fibre with various concentrations of 2%, 4%, and 6% for 24 h. The filtered PEFB suspension was acidified with 5 N H_2_SO_4_ until pH was 2. Next, acidified lignin was washed with distilled water and centrifuged at 8000 rpm for 5 min. The lignin was dried in a vacuum oven at 50 °C for 4 h. The products were coded as lignin_NaOH2%, lignin_NaoH4%, and lignin_NaOH6% for lignin obtained from PEFB fibre immersed in 2%, 4% and 6% NaOH, respectively.

### 2.4. Characterisation

#### 2.4.1. Scanning Electron Microscopy

Morphology of the samples was analysed by using a scanning electron microscopy (SEM, Hitacho TM3030, JEOL, Ltd., Tokyo, Japan) operating at 20 kV. The sample was first coated with a thin layer of gold before analysis to reduce charges during analysis.

#### 2.4.2. Transmission Electron Microscopy

A 0.01 CNF *w*/*v*% in distilled water was dispersed using a homogeniser. The suspension was then deposited on carbon-coated electron microscope grids and stained negatively with a drop of uranyl acetate. The sample was allowed to dry at room temperature before analysis. After that, it was observed with a transmission electron microscopy JEOL/EO JEM-1400 (TEM, JEOL, Ltd, Tokyo, Japan) operated at an accelerating voltage of 100 kV.

#### 2.4.3. Fourier Transform Infra-Red Characterisation

The functional groups in CNF and lignin were investigated using a FTIR Spectrometer (FTIR, Nicolet 380, Thermo Scientific, Boston, MA, USA). The samples for FTIR were prepared using KBr and a sample with comparison of 100:1. The instrument was operated in transmission mode with a wavelength of 400–4000 cm^−1^, resolution of 4 cm^−1^, and 50 scans.

#### 2.4.4. X-ray Diffraction Analysis

X-ray diffraction (XRD, Bruker D8 advanced X-ray diffractometer, Bruker Optic GmbH, Ettlingen, Germany), patterns were investigated using a Shimadzu XRD-6100 diffractometer with Cu-Kα radiation (λ = 0.154 nm) at scanning rate of 2°/min, 40 kV voltage and 200 mA current. The crystallinity of CNF was determined based on the Segal method.

#### 2.4.5. Particle Size Distribution

Particle size distributions of commercial and isolated lignin were analysed using a particle size analyser (LA-910, Horiba LA-910, Horiba Ltd, Kyoto, Japan). A diluted lignin solution was dispersed in distilled water and sonicated for 30 min. A few drops of the suspension were introduced to the dispersion unit in the instrument. The data were acquired over five cycles.

#### 2.4.6. UV-Visible Spectroscopy

UV-visible spectra were recorded on an ultraviolet/visible spectrophotometer (UV 1800 series, Shimadzu Scientific Instrument, Kyoto, Japan). A 5 mg of sample was dissolved in 0.1 mol/L NaOH solution (10 mL). One millilitre of the aliquot was diluted to 10 mL with distilled water and the absorbance in between 250 and 400 nm wavelength was measured. 

#### 2.4.7. ^1^H-NMR

^1^H-NMR spectra were recorded on an Agilent 500 MHz (NMR, Agilent Technology, Santa Clara, CA, USA) at a frequency of 500 MHz at room temperature with an acquisition time of 0.011.

#### 2.4.8. Thermogravimetric Analysis 

The mass-loss property of the materials was characterised using a thermogravimetric analyser (DTA/TG Exsstar SII 7300, Hitachi medical system, Tokyo, Japan). The sample was heated from 25 to 600 °C with a heating rate of 10 °C/minute under nitrogen atmosphere.

## 3. Results

### 3.1. Morphological Properties of Cellulose Nanofibre and Lignin

Figure 2a shows the morphology of CNF with 100× magnification. CNF morphology is seen as single fibres sticking to one another. This is probably due to hemicellulose, lignin, and pectin removal during chemical treatment. In addition, Figure 3 shows that the fibre has dimensions between 50 and 100 nm, which is calculated using ImageJ analysis. It is also seen that the CNF looks like whiskers. The results indicated that the CNF obtained has a nanoscale [25]. Another study reported that CNF isolated from a pineapple leaf using steam explosion had an interconnected web-like structure with a width between 5 nm and 60 nm [26]. The morphology of lignin precipitated with various NaOH concentrations is shown in Figure 2b–j. The morphology of lignin_NaOH2%, lignin_NaOH4%, and lignin_NaOH6% is observed as rough and flaky, whereas commercial lignin was in powder form with near-spherical morphology. The differences between isolated and commercial lignin morphology are present because of the different procedures used in manufacturing commercial lignin and lignin extracted in this study. From 1000 times magnification, there were slight pores on the surface of lignin_NaOH6% which were not observed in other isolated lignins. The presence of pores enhanced the dispersion of lignin with other polymers since they promote penetration of polymer into lignin molecules [27]. 

### 3.2. FTIR and UV Spectra

CNF and lignin FTIR spectral analyses were done to identify the functional groups present in the materials. FTIR spectra for these polymers are shown in Figure 4. 

With reference to Figure 4, the CNF absorption band at 3359 cm^−1^ represented hydrogen-bonded (O-H) stretching and C-H stretching was shown at 2900 cm^−1^. The presence of O-H bending that absorbed water was seen at 1600 cm^−1^. The absorption bands at 1371 cm^−1^ and 1200 cm^−1^ represented C-H in -O(C=O)-CH_3_ and C-O stretching of acetyl group in cellulose, respectively [28,29]. Both isolated and commercial lignin showed absorption bands at 1600 cm^−1^, 1515 cm^−1^, and 1425 cm^−1^ corresponding to aromatic ring vibrations of the phenylpropene (C9) skeleton [30]. Hydroxyl groups in aromatic and aliphatic structures were seen in a wide absorption range between 3600 and 3000 cm^−1^. C-H stretching in methyl and methylene groups of the side chain appeared at 2938 cm^−1^ absorption for all lignin. The peak at 2885 cm^−1^ corresponded to C-H stretching in aromatic methoxy group [31]. The attribution of the absorption peak for CNF and lignin is summarised in Table 1. 

UV spectroscopy was attributed to investigate lignin variability distribution and purity. Lignin consists of several functional chemical groups such as methoxy, carbonyl, carboxyl, and hydroxyl (phenolic and alcoholic). However, these groups are dependent on its original resources and the isolation procedure applied. Figure 5 shows UV absorption spectra of commercial and isolated lignin (lignin_NaOH2%, lignin_NaOH4% and lignin_NaOH6%). As seen on Figure 4, there was no significant difference within the acquired UV spectra for all lignin samples. All lignin samples had maximum absorption at around 280 nm which indicated the presence of aromatic rings/non-conjugated phenolic groups in the lignin structure [34,35]. In addition, UV-vis spectra for CNF dispersion in deionised water showed optimum absorption at a wavelength of ~280 nm.

### 3.3. ^1^H-NMR of Pure Lignin

The chemical structures of isolated and commercial lignin were analysed using ^1^H-NMR spectrometry. The lignin sample ^1^H-NMR result is presented in Figure 6. Protons in phenolic hydroxyl can be seen in the signals between 8.0 and 9.0 ppm. The integration of signals in the 6.2–7.6 ppm range could be attributed to aromatic and vinyl protons in syringyl (S) and guaiacyl (G) units. Aliphatic protons appeared in the signals from 4.2 to 5.6 ppm. The signal at around 3.8 ppm was related to the proportion of the G and S units. The sharp signals at 3.4 ppm and 2.5 ppm were attributed to a water contaminant and DMSO-d6 solvent, respectively. Water contaminants appeared in both isolated and commercial lignin samples, as the sample was not well-dried before analysis; meanwhile, the DMSO-d6 used was not 100% pure. Finally, protons in saturated aliphatics were seen in the 2.3–1.2 ppm range.

### 3.4. X-Ray Diffraction Analysis

X-ray diffraction (XRD) analysis was performed to investigate the lignin properties in more detail as well as the crystallinity of CNF. Isolated and commercial lignin showed amorphous patterns (Figure 7). From Figure 7, a crystallographic plane of native cellulose structure was assigned at the peak range between 21.90° and 22.20° 2θ. All of the diffractogram is described in more detail as follows: 14.5°–15.3° 2θ reflection attributed to the (1–10) crystallographic plane, the 15.7°–16.30° 2θ reflection attributed to the (110) crystallographic plane, the 18.30°–18.40° reflection attributed to the amorphous phase, and the 21.90°–22.20° 2θ reflection attributed to the (200) crystallographic plane of cellulose I [29,36]. Based on the Segal equation, the crystallinity index of the CNF was calculated at 72%.

### 3.5. Particle Size Analysis of Lignin

The particle size analysis of lignin is shown in Figure 8. Particle size distribution of commercial lignin was between 1 and 300 μm, which is almost the same as lignin_NaOH2% and lignin_NaOH4%, while lignin_NaOH6% had the smallest particle size, from 1–30 μm. At the sample preparation step, lignin was first dispersed in deionised water and sonicated using a homogeniser at 300 rpm with various durations. Lignin_NaOH6% was sonicated for the longest time, hence the slight shift of its distribution to lower sizes. The large particle size of lignin may also be correlated to poor lignin solution in common organic solvent. It was found that lignin solution precipitated when it was stored for a longer time. 

### 3.6. Thermogravimetric Analysis

Thermogravimetric analysis (TGA) was done to analyse the thermal stability of CNF, commercial lignin, and isolated lignin. The thermal characteristic results are summarised in Table 2, while the thermogravimetric traces for lignin and CNF are presented in Figure 9. 

Thermogravimetric traces showed thermal degradation temperatures which were attributed to samples weight loss. In Figure 8, the lignin TGA curve showed a three-stage processing time function. First, the dehydration of samples started from room temperature to 100 °C. There was 1%–3% weight loss for isolated lignin and 4% for commercial lignin. At this stage, the moisture content in lignin was removed. Second, the thermal degradation of lignin samples was in the temperature range of 225 to 350 °C. There was almost 50% weight loss for isolated lignin, and the most prominent trend is seen for lignin_NaOH6%. However, the mass decrease for commercial lignin at this step was only approximately 25%. Finally, lignin samples lost almost all of their weight at 600 °C. There were remaining residual masses of 3.5%, 0.8%, 1.7%, and 1.9% for lignin_NaOH2%, lignin_NaOH4%, lignin_NaOH6%, and commercial lignin, respectively. 

The TGA trace for CNF also shows three distinct regions. The first one was between room temperature and 100 °C where CNF was dehydrated. The second region was observed from 270 to 400 °C where there was decomposition of CNF to its monomers, D-glucopyranose. Last, decomposition was completed at 600 °C with approximately 30% carbon yield, which was higher than that of lignin. From the data, thermal decomposition of the CNF was higher than that of lignin. Increasing of the crystallinity index of material shifted the thermal decomposition to a higher temperature [37]. The same value also has been reported from different raw materials, such as Alfa fibre [38] and wood [39]. The crystalline regions of CNF act as barriers for the heat transfer, hence they improve the thermal stability of CNF [36]. 

## 4. Conclusions

Cellulose nanofibre (CNF) and lignin from palm empty fruit bunches was successfully isolated without by-product. In this study, the isolation of CNF was achieved successfully using steam explosion and lignin using soda pulping methods. It was shown that the method used is effective and efficient. CNFs have diameters between 50 and 100 nm with 72% crystallinity index. Commercial lignin was in powder form with near-spherical morphology, whereas isolated lignin has flaky and rough morphology. Particle size distributions of commercial and isolated lignin are almost the same, between 1 and 300 μm. From UV-visible analysis, lignin has aromatic rings/non-conjugated phenolic groups. Thermal properties demonstrated that CNF has higher thermal stability than lignin associated with high crystallinity of CNF (72%). It can be concluded that lignin and CNF of PEFB have the potential to be used as precursors for carbon fibre and reinforcement materials.

## Figures and Tables

**Figure 1 materials-13-02290-f001:**
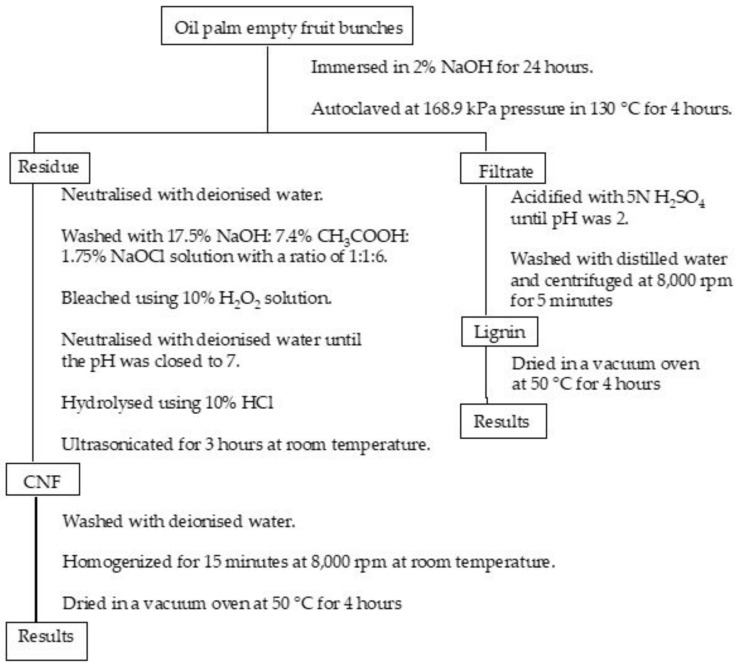
A flowchart of isolating cellulose nanofibre and lignin from oil palm empty fruit bunches.

**Figure 2 materials-13-02290-f002:**
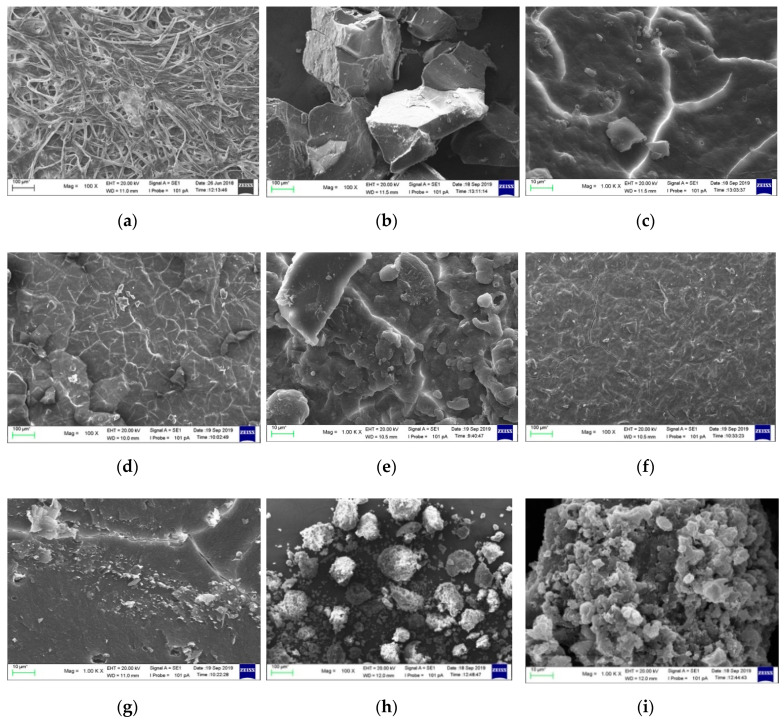
The morphology of: (**a**) cellulose nanofibre (CNF), (**b**,**c**) lignin_NaOH2%, (**d**,**e**) lignin_NaOH4%, (**f**,**g**) lignin_NaOH6%, and (**h**,**i**) commercial lignin.

**Figure 3 materials-13-02290-f003:**
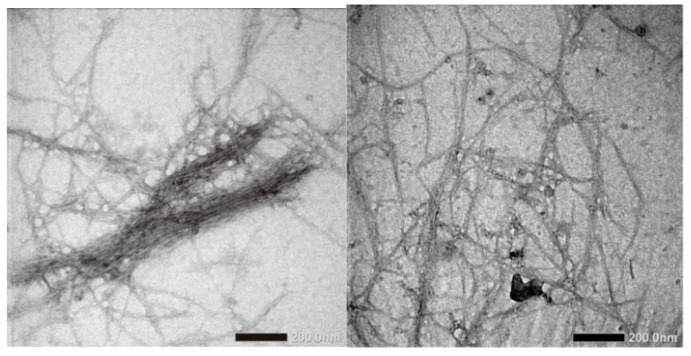
TEM images of CNF from palm empty fruit bunches produced by steam explosion.

**Figure 4 materials-13-02290-f004:**
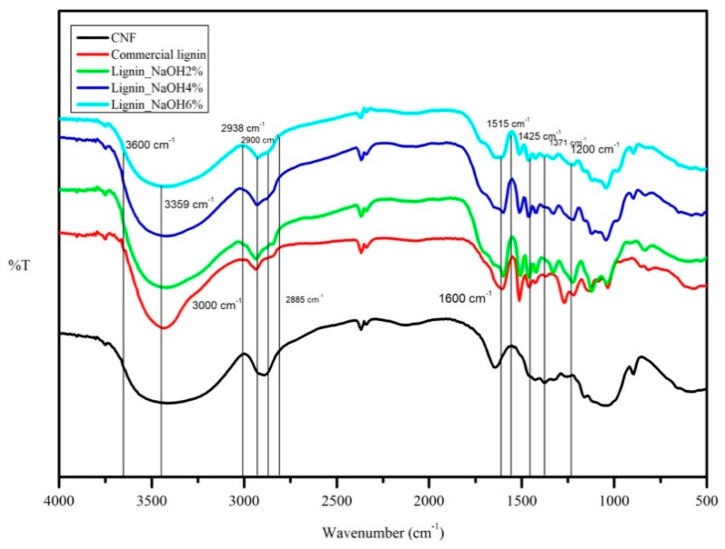
FTIR spectra of CNF, lignin_NaOH2%, lignin_NaOH4%, lignin_NaOH6%, and commercial lignin.

**Figure 5 materials-13-02290-f005:**
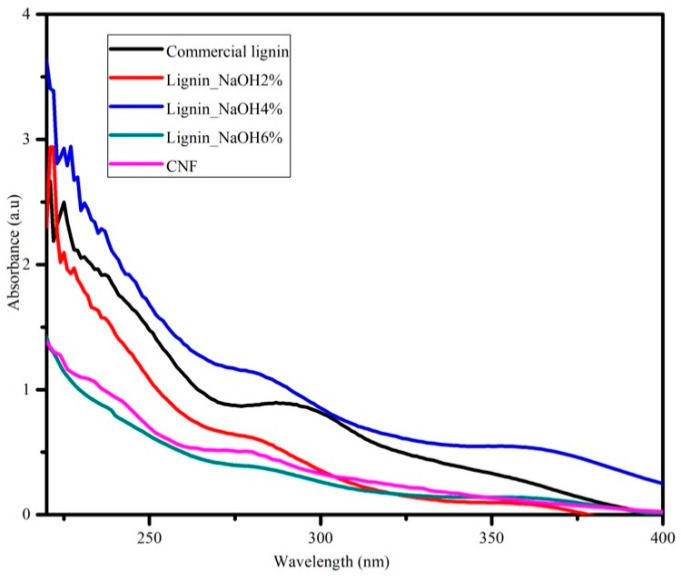
Ultraviolet-visible (UV-vis) analysis of commercial lignin, lignin_NaOH2%, lignin_NaOH4%, lignin_NaOH6%, and CNF.

**Figure 6 materials-13-02290-f006:**
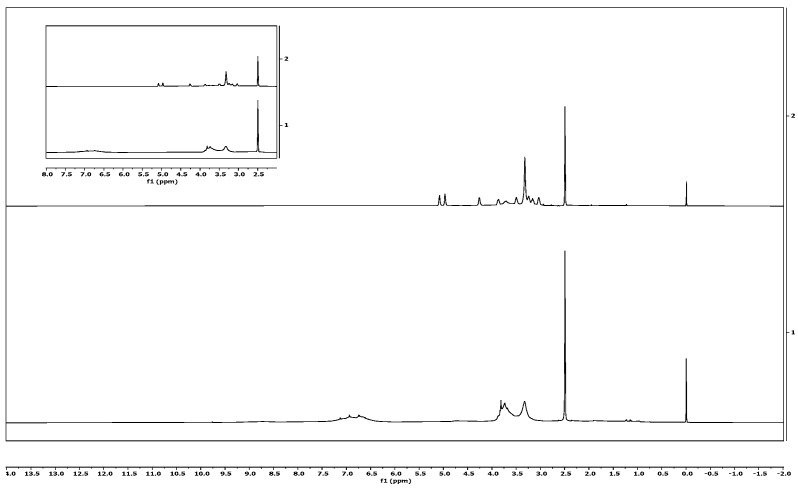
Signals of proton NMR of isolated lignin (above) and commercial lignin (below).

**Figure 7 materials-13-02290-f007:**
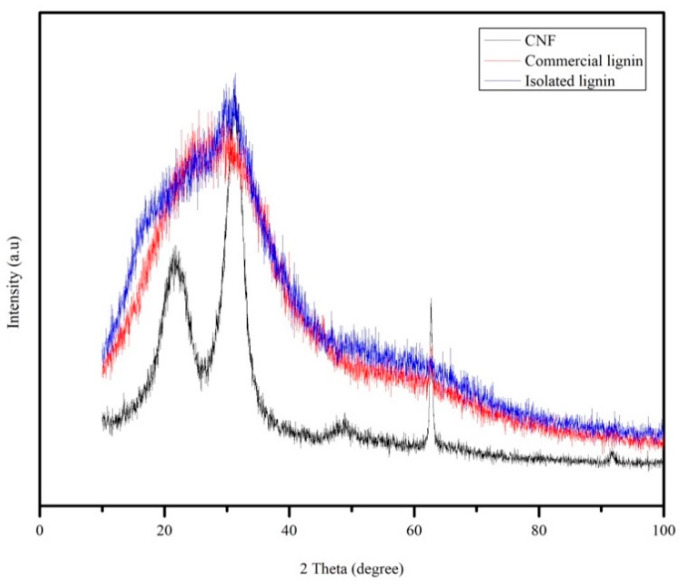
XRD pattern of CNF, commercial, and isolated lignin.

**Figure 8 materials-13-02290-f008:**
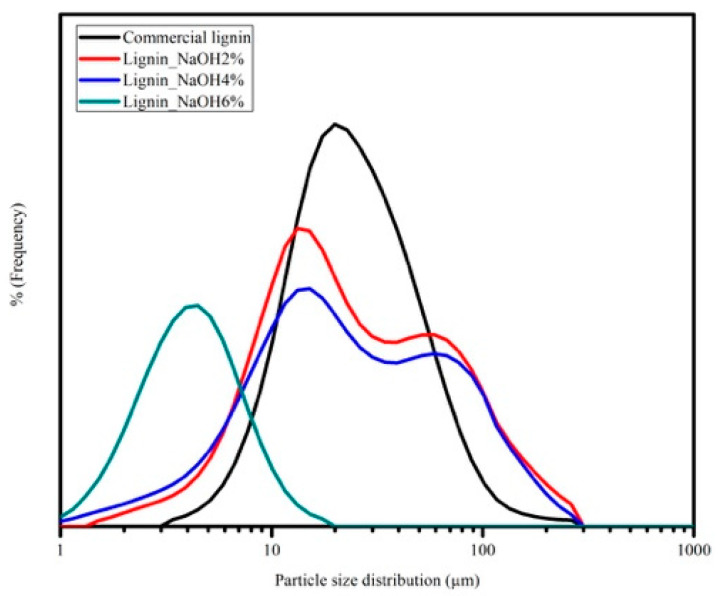
Particle size distribution of commercial lignin, lignin_NaOH2%, lignin_NaOH4%, and lignin_NaOH6%.

**Figure 9 materials-13-02290-f009:**
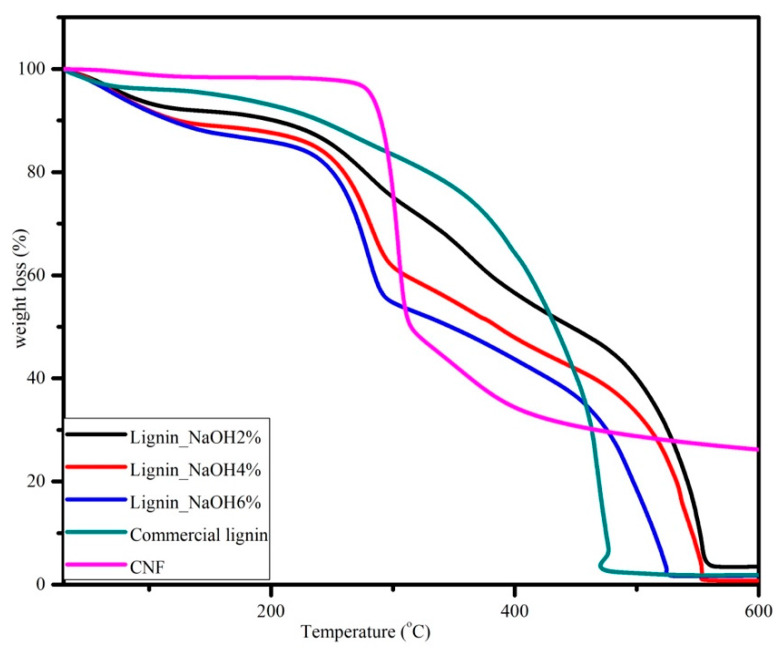
Thermogravimetric analysis curves for CNF, commercial, and isolated lignin with 10 °C/minute heating rate.

**Table 1 materials-13-02290-t001:** FTIR spectral assignments for CNF and lignin [30,32,33].

Wavenumber (cm^−1^)	Attribution to the Band CNF Lignin	
3430		O-H stretching of hydroxyl groups
3278-3495	O-H stretching	
2940-2840		C-H stretching of aliphatic and aromatic structure
2890	H-C-H stretching (alkyl, aliphatic)	
1765-1705		C=O stretching of carboxylic groups
1720-1740	C=O stretching (carbonyl)	
1640	Fibre-OH (absorbed water)	
1600		C-C stretching of aromatic skeleton
1611,1517,1425		Vibrations of aromatic rings
1429	HCH and OCH bending vibration (methylene group)	
1371,1200	CH deformation vibration (CH_3_ or OH in plane)	
1328, 1271, 1114		Vibrations of C-H bonds in aromatic rings
1270-1232	C-O-C ether bond (aryl-alkyl ether)	
1220		C-O(H) + C-O(Ar) stretching
1170-1082	C-O-C antisymmetric bridge stretching (Pyranose ring skeletal)	
1108	OH (C-OH)	
1071-1067	C-O stretching	
1041-1054	C-O symmetric stretching (C-O of primary alcohol)	
896-915	COC, CCO, and CCH deformation and stretching vibration	
834		Vibrations of C-H bonds in aromatic rings
700	CH_2_ vibrations	

**Table 2 materials-13-02290-t002:** Thermal characteristic of CNF and lignin samples.

Sample	First Decomposition		Second Decomposition		Third Decomposition	
	Residual Mass (%)	T_max_ (°C)	Residual Mass (%)	T_max_ (°C)	Residual Mass (%)	T_max_ (°C)
Lignin_NaOH2%	93.0	105.7	66.4	350.6	3.5	600.5
Lignin_NaOH4%	91.4	105.6	54.9	350.4	0.8	600.8
Lignin_NaOH6%	91.2	105.5	49.6	350.7	1.7	600.6
Commercial lignin	96.1	105.3	76.9	350.9	1.9	600.5
NFC	99.8	105.8	35.5	400.0	30.2	600.0

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
