# Peer review of "Isolation and Characterisation of Cellulose Nanofibre and Lignin from Oil Palm Empty Fruit Bunches"

_materials, 2020, doi:10.3390/ma13102290_

Round 1
Reviewer 1 Report
1) The title (one-step isolation) is misleading. Title change required
2) The extraction method used to extract cellulose is adopted from previous work. It is important that the authors say that (currently, this is self-referenced as [14] which need to be changed)
3) Authors have presented part of this work in a conference and extraction part of the work is published ("Isolation of nanofibre cellulose from oil palm empty fruit bunch via steam explosion and hydrolysis with Hcl 10%"). It is important to mention.
4) This article is very similar to other published articles. The novelty I find is the their "Oil palm fruit bunches". It is important to establish the novelty.
Reviewer 2 Report
The manuscript is well written, clear and concise. It describes a simple process to extract cellulose nanofiber and lignin from oil palm empty fruit bunch. It needs some corrections to be accepted with minor revision. These are listed below.
- Line 13. “NFC was extracted by steam explosion …”
- Line 20. “The two lignins …”
- Line 32. “biodegradability”
- Lines 43-44. “Lignin … as the second most abundant natural resource after cellulose.” This is a questionable statement. Please include a reference.
- Line 51. “… low density and has good mechanical properties.”
- Line 56. “Research and Development”
- Lines 61-62. “… the lack of carbon fibre from lignin to modulus and strength can be increased.” This portion of the sentence makes no sense. Please rewrite.
- Line 64. “… lignin and cellulose …”
- Lines 66-67. “The utilization of PEFB as a source to produce valuable materials … as the components of this polymer have the potential to be an alternative to renewable energy.” In this sentence, when you say that PEFB can be a source of energy you mean that it can produce fuel? If so, the components of PEFB and not “polymer” can be an alternative renewable energy.
- Line 87. “… was close to 7.”
- Line 88. “… and ultrasonicated for …”
- Line 97. “… for lignin obtained from PEFB fibre immersed in 2%, …”
- Lines 110-111. What do you mean with “… sample disk with comparison of 100:1.”?
- Line 115 and beyond. The “degree” symbol has an unneeded underline. Please correct at all places concerned.
- Line 123. “UV-visible spectra were …”
- Line 137. “… is seen as single fibres sticking to each other.”
- Line 144 “… differences between isolated and commercial lignin …”
- Line 148. “… they promote …”
- Figure 1(b). Judging from the 10-micrometer scale bar, this picture appears to show fibres and not NFC as in Fig 1(a), but the caption says “… (a-b) NFC, …”. Still in the same caption, the other images are referred as being of lignin1, lignin2, lignin3. Please make it consistent with the rest of the text (lignin_NaOH?%).
- Lines 174-175 describes the band at 1200 cm-1 as corresponding to C-O stretching of acetyl group in cellulose, but there is no attribution at this wavelength to cellulose in Table 1.
- Line 192. “… groups in the lignin structure …”
- Line 210. “performed”
- Line 213. “The result was strengthen from FTIR spectrum and XRD analysis carried out.” The sentence does not make sense. Please rewrite.
- Line 220. Figure 7 shows particle size analysis of lignins only, not NFC as written in this line. Figure caption also mentions NFC but there are no NFC size distribution curve.
- Line 235. “… which were attributed …”
- Line 237. “dehydration” as in line 247 (“dehydrated”).
- Line 242. There is an f in “… weight at f 600°C”
- Caption of Fig. 8. “… curves for NFC, commercial and isolated lignin …”
Reviewer 3 Report
Manuscript: One-step Isolation of Cellulose Nanofibre and Lignin from Oil Palm Empty Fruit Bunches
Materials-776520
Manuscript offered very good research connected to removal of Naocellulose and can be accepted for publication after minor change.
- Author need to include some important data in the abstract part of the manuscript
- Author need to include some recent reference related to nanocellulose in the introduction part to make it stronger.
- Authors need to Improve quality of figure all figures in the manuscript.
- Please indicate important peak in FT-IR.
- Page 3, Line 96 correct o
- What is the % of lignin present in the samples?
- What is the peak in FT-IR around 1600 Cm- ?
- Please indicate all the planes in PXRD.
- What is the % yield for nanocellulose?
- Why authors method to remove nanocellulose is superior from other reported methods?
Reviewer 4 Report
Manuscript: One-step Isolation of Cellulose Nanofibre and Lignin from Oil Palm Empty Fruit Bunches
Manuscript presented very good research related to separation of cellulose Nanoforms and recommended for publication after minor change.
- Introduction part look poor: authors need to put more information in that related to subject.
- Page 5, Please label important peaks in FTIR.
- Improve quality of figure 5.
- Improve quality of figure 6 with proper peak labelling.
- What is the residual weight in TGA profile, please explain?
Reviewer 5 Report
The authors reported a study of separating lignin and cellulose nanofibre from oil palm in a one-step procedure. Overall the experiments were well-conducted with good presentation. I suggest for publications with below questions being addressed.
- The authors isolated lignin and NFC in a one-step protocol, but what is the major improvements and differences compared with current procedures like kraft or soda process? It will be helpful to illustrate the common industrial separation processes in introduction section. Also, a flow chart will be a straightforward way to show your process.
- The authors mentioned carbon fiber as one of the significant application for lignin. A few highly impact papers should be cited as listed below,
- Baker, D.A., Gallego, N.C. and Baker, F.S. (2012), On the characterization and spinning of an organic‐purified lignin toward the manufacture of low‐cost carbon fiber. J. Appl. Polym. Sci., 124: 227-234.
- Meng Zhang, Amod A. Ogale, Carbon fibers from dry-spinning of acetylated softwood kraft lignin, Carbon, Volume 69, 2014, Pages 626-629
- One of the properties that could significantly affect the quality of lignin derived carbon fiber is ash. If the carbon fiber application is addressed in the introduction part, an ash content study should be performed using either muffle oven or TGA in air.
- Page 2 line 64 a typo of “dan” should be “and”.
Reviewer 6 Report
The objectives of the paper are clear and well described however the manuscript is more a technical report than a scientific study. Furthermore the novelty of the work and its potential applicability is not clear. The poorest part is related to the nanocellulose (NC) part. Unfortunatly, I cannot recomend the publication of this manuscript.
INTRODUCTION
Need to be focused on the use of alternative biomass to produce NC and lignin. That means that the relevance of NC and lignin should be briefly mentioned with a couple of general review references (e.g. Nanocellulose for industrial use: cellulose nanofibers (CNF), cellulose nanocrystals (CNC), and bacterial cellulose (BC)2018. A Blanco, MC Monte, C Campano, A Balea, N Merayo, C Negro. Handbook of Nanomaterials for Industrial Applications, 74-126; Industrial Application of Nanocelluloses in Papermaking: A Review of Challenges, Technical Solutions, and Market Perspectives 2020. A Balea, MC Monte, N Merayo, C Campano, C Negro, A Blanco. Molecules 25 (3), 526; Nanocelluloses: natural-based materials for fiber-reinforced cement composites. A critical review. 2019. A Balea, A Blanco, C Negro. Polymers 11 (3), 518) and then focus on the use of residual biomass for NC and lignin production. This has been already done and it should be referenced; furthermore the results of the manuscript should be compared with published similar data. There are many examples for NC e.g. Cellulose nanofibers from residues to improve linting and mechanical properties of recycled paper A Balea, N Merayo, E Fuente, C Negro, M Delgado-Aguilar, P Mutje, ...Cellulose 25 (2), 1339–1351; Valorization of corn stalk by the production of cellulose nanofibers to improve recycled paper properties 2016. A Balea, N Merayo, E Fuente, M Delgado-Aguilar, P Mutje, A Blanco, ...BioResources 11 (2), 3416-3431); the same should be done for the state of the art of obtaining lignin from residual biomass.
The potential of this kind of residues should be quantify to explore their real industrial application.
RESULTS:
NC need to be compared with the properties of similar NC products obtained from other residues and wood resources in a similar way authors have compared the lignin results with commercial lignin.
The scientific discussion of the data is really poor.
CONCLUSIONS:
The important is the potential applicability and the comparison with other similar materials.
Round 2
Reviewer 6 Report
The poorest part of this paper which still contains significant scientific mistakes is related to the nanocellulose (NC). Unfortunatly, I cannot recomend the publication of this manuscript as it is because the conceptual mistakes on this part. However, if the authors are able to correct them the rest of the manuscript is fine.
The main problem is the confusion between cellulose nanofibres (CNF) and cellulose nanocrystals (CNC). Authors should follow the ISO standard (ISO/TS 20477:2017:Nanotechnologies -- Standard terms and their definition for cellulose nanomaterial). This standard reviewed all nomenclature used for these NC and has standardized their use. Therefore the abreviatures used in the manuscript are not correct.
The sentence in the abstract "The nanowhisker of NFC with dimension between 50 nm and 100 nm was investigated" is very confussing because nanowhiskers are not nanofibres and in this case they were not obtained from nanofibres but from cellulose directly. Furthermore nanowhiskers correspond to an old nomenclature as already mentioned. They should speak about nanocrystals.
Due to this issue the references are confuse. Authors give references of nanocrystals while speaking about nanofibres. I am not sure when they would like to consider one or the other.
I have made some suggestions on the references but authors should define clearly what they are speaking about in each moment.
Conclusions are confuse due to the use of NCF and CNC nearly as sinonimus.
I have made a few comments and suggestions on the references in the attached file.
Since authors have not marked the changes in a different colour it is difficult to follow the improvements of the version 2 of the manuscript.

Author Response
The term for cellulose nanofibre as CNF has been changed in all the manuscript. It has been followed regarding ISO/TS 20477:2017:Nanotechnologies.
The word of nanowhisker has been replaced.
Reference of number 9 has been replaced to be more appropriate reference regarding to the suggestions references.
The word of nanowhiker in the conclusion has been replaced, the same as in the abstract.